# Rapid poxvirus engineering using CRISPR/Cas9 as a selection tool

Anjali Gowripalan [1,2], Stewart Smith[1,2], Tijana Stefanovic[1] & David C. Tscharke [1✉]

In standard uses of CRISPR/Cas9 technology, the cutting of genomes and their efficient repair are considered to go hand-in-hand to achieve desired genetic changes. This includes the current approach for engineering genomes of large dsDNA viruses. However, for poxviruses we show that Cas9-guide RNA complexes cut viral genomes soon after their entry into cells, but repair of these breaks is inefficient. As a result, Cas9 targeting makes only modest, if any, improvements to basal rates of homologous recombination between repair constructs and poxvirus genomes. Instead, Cas9 cleavage leads to inhibition of poxvirus DNA replication thereby suppressing virus spread in culture. This unexpected outcome allows Cas9 to be used as a powerful tool for selecting conventionally generated poxvirus recombinants, which are otherwise impossible to separate from a large background of parental virus without the use of marker genes. This application of CRISPR/Cas9 greatly speeds up the generation of poxvirus-based vaccines, making this platform considerably more attractive in the context of personalised cancer vaccines and emerging disease outbreaks.

[1] John Curtin School of Medical Research, Australian National University, 131 Garran Road, Acton, ACT 2601, Australia. [2] These authors contributed equally: Anjali Gowripalan, Stewart Smith. ✉email: david.tscharke@anu.edu.au

The ability to edit genomes quickly and efficiently has been accelerating within recent years due to the availability of novel genome editing tools. Most notably, CRISPR/Cas9 has been adapted to serve as a robust platform for generating recombinant mammalian, bacterial, fungal, protist and viral genomes within the laboratory[1–5]. In virology, CRISPR/Cas9 editing has proven useful in engineering medium to large DNA viruses, such as adenoviruses, herpesviruses and African swine fever virus[6–8]. In these cases, CRISPR/Cas9 greatly facilitates homologous recombination with transfer plasmids, which otherwise occurs at frequencies too low for identification of recombinants without mass screening or a means of selection[5]. The fundamental mechanism of CRISPR-mediated genome editing is two-fold: First, DNA is cleaved at a specified point by Cas9-guide RNA complexes (Cas9/gRNA) and second, these double-stranded breaks (DSBs) are efficiently repaired by cellular enzymes. The repair can be achieved by non-homologous end joining (NHEJ), which is a non-templated mechanism that can incorporate indels, or homology-directed repair (HDR), if a suitable template is available[9–11].

Vaccinia virus (VACV) is a large, dsDNA virus in the *Orthopoxviridae* family. It is most renowned for its use in the eradication of smallpox, but continues to serve as a vector for recombinant vaccines and is an agent for oncolytic virotherapies[12,13]. One particularly desirable feature of VACV is the amenability of the 200 kb genome to insertions and deletions of up to 25 kb[14]. This flexibility allows for the introduction of genes for multiple antigens and immune-modulating proteins. In addition, ongoing studies are identifying viral genes that can be removed to increase safety and/or immunogenicity[15–17]. VACV, like all poxviruses, replicates within the cytoplasm of infected cells eliminating the possibility of integration into the host genome, which further supports its safety as a vector[18]. The replication cycle is initiated by uncoating of viral cores in the cytosol and genome replication proceeds in membrane-encased structures known as virus factories using a viral polymerase[19,20]. These perinuclear sites segregate VACV genomes from the cell nucleus, and are also a key site for homologous recombination[21,22]. In fact, replication of the VACV genome is reliant on efficient recombination[23]. This is due to the non-continuous nature of VACV DNA replication which produces genomic breaks at replication forks, forcing VACV to carry its own machinery for repair at these sites[23–28]. Due to this feature, genome editing of VACV by homologous recombination is relatively efficient compared with other DNA viruses, such as herpesviruses, which rely on an array of host repair enzymes[6,29]. However, this efficiency remains in the range of 0.1–1% for VACV, meaning marker genes are inevitably required to enable selection or screening of desired recombinants from a large background of unmodified parent virus. Currently, marker-free recombinants require time-consuming and labour-intensive methods, such as transient dominant strategies[30]. The application of CRISPR/Cas9 to facilitate engineering of VACV has been reported, but despite the first publication 5 years ago, this method has not gained widespread use in the field[31–33].

Here we show that while VACV genome cutting is efficiently achieved by CRISPR/Cas9, repair by NHEJ or HDR is not efficient. This means that CRISPR/Cas9 is not a reliable tool for increasing rates of homologous recombination. Instead, inefficient repair leaves the cut genomes unable to be replicated. With this insight we go on to show that the utility of CRISPR/Cas9 lies in its use as a highly versatile and efficient selection method for recombinant VACVs, allowing for generation of marker-free viruses in under 2 weeks.

## Results

**Low efficiency of NHEJ repair following CRISPR/Cas9 targeting of VACV.** We have previously described a successful protocol for editing of herpes simplex virus (HSV) genomes with CRISPR/Cas9[34–36]. Despite having this system in place, our parallel experiments with VACV were met with little success. For this reason, we set up fluorescent virus systems to quantify the efficiency of any CRISPR/Cas9 activity, starting without a repair template to examine VACV genome editing by NHEJ. To ensure we would have active Cas9/gRNA in the cytoplasm, we transfected these as ribonucleoprotein (RNP) complexes instead of expressing them in cells from plasmids. We infected cells with a VACV expressing mCherry from a disrupted viral thymidine kinase (TK) locus and transfected Cas9 complexes loaded with either an mCherry (mCh)-specific or TK-specific gRNA as test and negative control, respectively (Fig. 1a). After 48 h, the level of red fluorescence appeared to be lowest for the culture treated with the mCh-gRNA by microscopy (Fig. 1b) a result that was supported by flow cytometry (Fig. 1c). The majority of the cells were then frozen and thawed to release the virus, serially diluted and used to infect new wells so that plaques could be scored for loss of fluorescence as evidence of genome editing (Fig. 1d). All plaques from control cultures were fluorescent, but non-fluorescent plaques appeared at ~5% when the mCh-gRNA was used (Fig. 1d). We noted that this low fraction of mCh⁻ VACV plaques did not match the roughly 30% reduction of fluorescence seen by flow cytometry (Fig. 1c). In similar experiments, including many using various plasmids to deliver Cas9/gRNA, this fraction of non-fluorescent plaques has ranged from <1% to as much as 10%, but never approached the routinely high levels (40–60%) seen in previous work when using the same gRNA to target an mCherry-expressing HSV[34]. A difference in protocol between our HSV and initial VACV experiments was the order of infection and transfection of cultures. However, when the order of these steps was compared directly for VACV we found the same level of editing irrespective of whether infection with parent virus or transfection of Cas9/gRNA was done first (Fig. 1e).

To next determine if inefficient gene editing by CRISPR via NHEJ was specific to the mCherry gRNA or the *TK* locus, we used the same protocol to knock out *eGFP* when located in three different sites of the VACV genome as fusions to genes *F5L, A3L* and *B5R* (Fig. 1f, Supplementary Fig. 1). When these viruses were used to infect cells transfected with Cas9/eGFP-gRNA complexes, green fluorescence was reduced (Supplementary Fig. 2), but, regardless of location, recovery of GFP⁻ plaques was low (Fig. 1g)[37,38]. Together with our experiments with mCherry, these data show that poor CRISPR editing of VACV genomes by NHEJ is a general finding across genome sites and target genes.

**CRISPR/Cas9 targeting does not improve recombination frequency for VACV.** NHEJ requires a range of host proteins, not all of which might be available in the cytoplasm. By contrast, VACV proteins are sufficient for homologous recombination. Accordingly, we went on to test whether Cas9 targeting might lead to efficient editing of VACV by HDR. In these experiments, we used a non-recombinant VACV (VACV WR), a repair plasmid with an mCherry expression cassette and homology arms matching sequences either side of the viral *TK* gene (pSC11-mCh), and a Cas9/TK-gRNA complex (Fig. 2a). Cells were infected with virus and simultaneously transfected with Cas9 complexes and the repair plasmid. After 48 h, virus was harvested and dilutions used to infect new cultures for the assessment of successful insertion of mCherry by microscopy (Fig. 2b). There was no significant difference in the proportion of mCh⁺ plaques

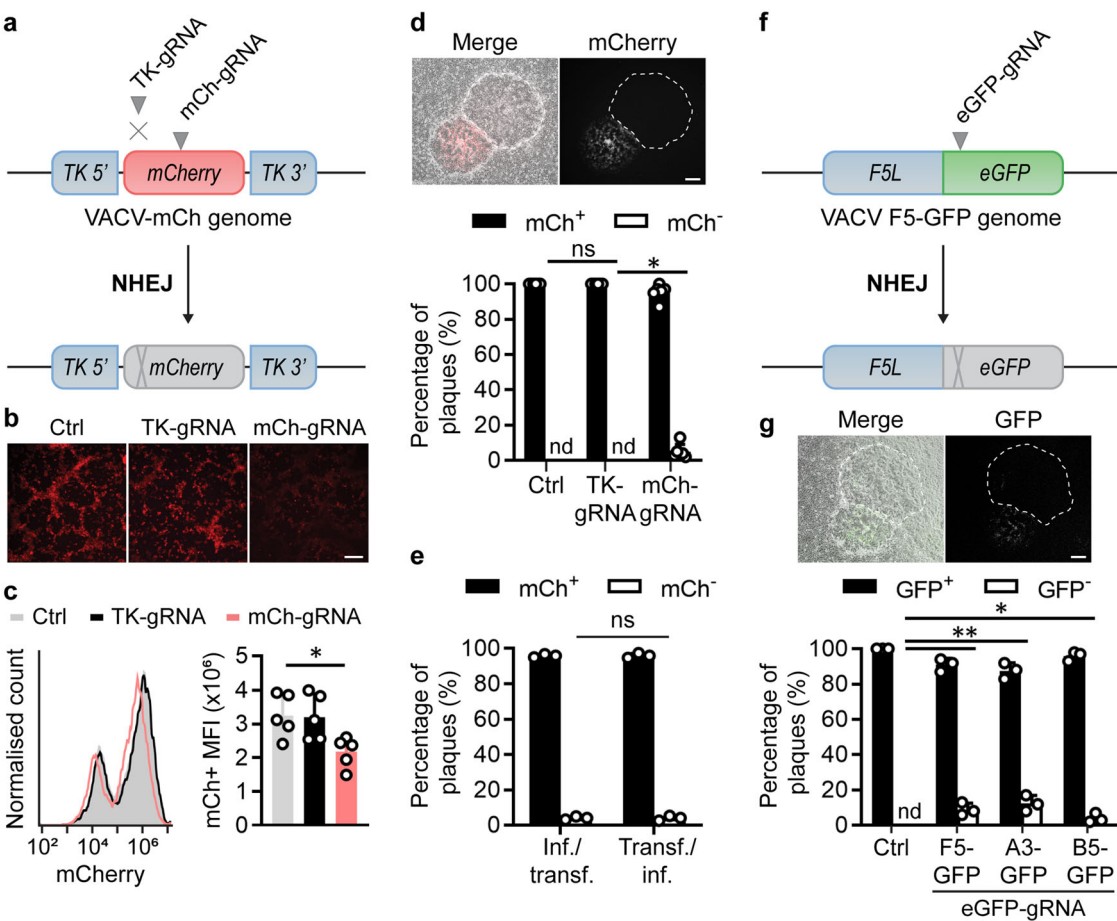

**Fig. 1 Inefficient mutagenesis of VACV by CRISPR/Cas9-induced NHEJ. a** Schematic showing genome of VACV-mCh and the location of gRNA target sites. **b**, **c** Fluorescence micrographs (**b**) and flow cytometry (**c**) of VACV cultures 48 hrs after infection with VACV-mCh and transfection with the Cas9/gRNA complexes or no gRNA (Ctrl). Scale bar on micrographs = 200 μm (**b**); representative histogram and means and SEM of 5 replicates shown for MFI (*$p$ = 0.0364 by one-way ANOVA) (**c**). **d** Progeny from cultures in (**b**) was grown on new monolayers and plaques scored for red fluorescence by microscopy. Example plaques are shown (scale bar = 300 μm). One hundred plaques from each culture across five independent experiments are graphed showing mean and SD (*$p$ = 0.0037 by two-way ANOVA). **e** Experiment described above was recapitulated with reversed order of infection and transfection. **f** Schematic showing genome of VACV F5-GFP and gRNA target site as an example of the three additional viruses used. **g** Fluorescent plaque scoring was conducted as above (*$p$ = 0.0150, **$p$ = 0.0011 by two-way ANOVA, ns = not significant, nd = not detected, $n$ = 3).

produced with or without the addition of the Cas9/TK-gRNA complexes. This suggests that targeting of Cas9 to the insertion site did little, if anything, to improve upon basal rates of homologous recombination between the VACV genome and HDR repair template.

To test if the results above were a function of the site targeted we investigated a second locus in the genome (Fig. 2c)[39]. In this experiment, the repair plasmid, pBII-ΔR, was used to replace 23 kb at the right arm of the VACV genome, from genes *Spi2* (*B13R*) to *C11R* in the inverted terminal repeat (ITR), with mCherry. Not only is this a different region of the genome, but the area being manipulated is much larger and so the generation of this virus might be considered more challenging. Nevertheless, the Cas/gRNA complex that we used to target a site in the *Spi2* gene that is deleted in the desired recombinant had no significant impact on the frequency with which mCh+ plaques were found (Fig. 2d). Indeed, recombination rates were similar as seen in our previous experiment. These data demonstrate that regardless of site or size of a genetic manipulation, the use of CRISPR does little to increase the frequency of genome editing by HDR for VACV.

**CRISPR/Cas9 impairs VACV replication**. In the NHEJ experiments above, as well as the substantial reduction of fluorescence

in Cas9/mCh-gRNA targeted cultures, we also noticed less cytopathic effect, hinting at reduced virus replication (Fig. 3a). These suggest that there was an effect of Cas9/gRNAs on VACV growth. For this reason, we formally quantified the infectious virus produced in each culture using a standard plaque assay. VACV-mCh production was significantly reduced in cultures transfected with Cas9/mCh-gRNA complexes compared with the irrelevant TK-gRNA or no Cas9 (Fig. 3b). This result was surprising because mCherry is a non-viral gene and so non-essential for VACV replication. This led us to ask whether targeting an essential VACV gene might impact viral replication to an even greater extent. We chose *A23R*, which (a) encodes a viral transcription factor, (b) is critical for viral replication and (c) has been targeted with CRISPR/Cas9 previously[33,40,41]. When *A23R* was targeted in VACV-mCh infected cultures, production of virus was reduced to the same extent as when mCh was the gRNA target (Fig. 3c). These data were again recapitulated when we targeted other sites in the genome by using Cas9/eGFP-gRNA complexes to target *eGFP* in VACV F5-, A3- and B5-GFP viruses (Fig. 3d). These results suggest that it does not matter if an essential or non-essential gene is targeted by Cas9/gRNA complexes, the outcome will be a similar substantial impairment to VACV replication. This is reminiscent of a study where cells treated with GFP- or

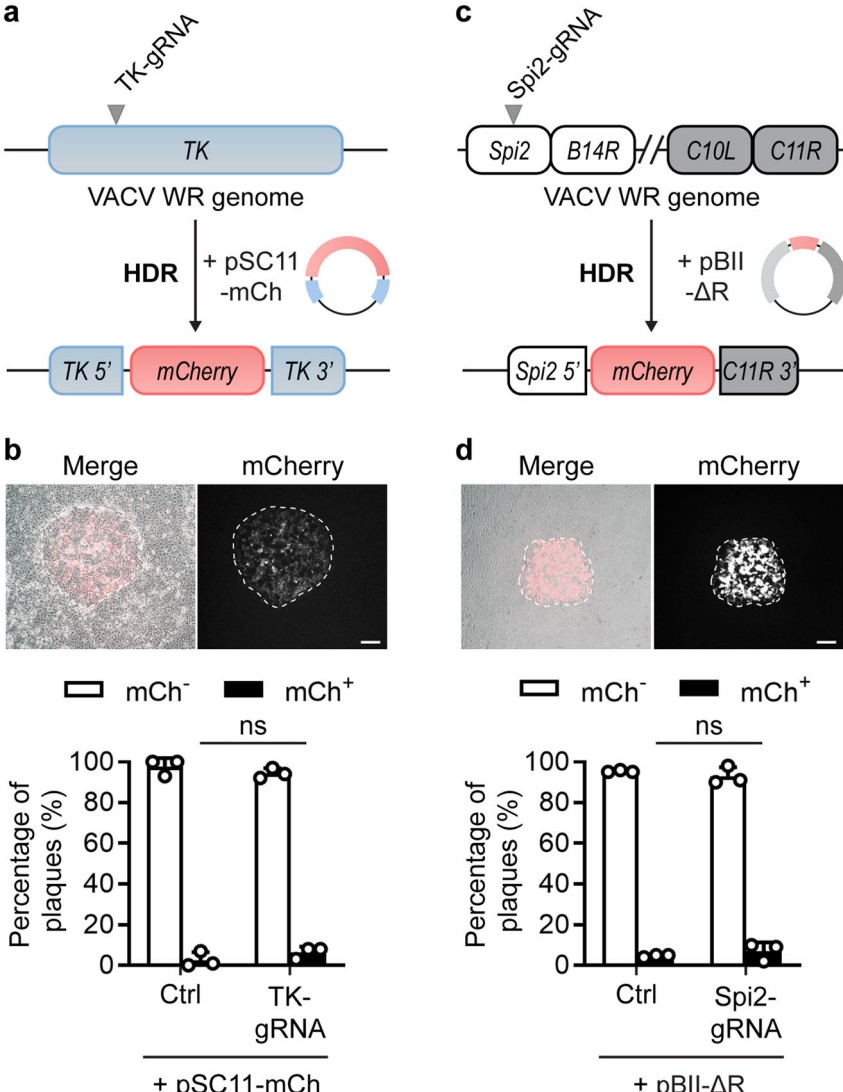

**Fig. 2 VACV Recombination is not improved by Cas9/gRNAs. a** Schematic of VACV genomes, gRNA target site and plasmids used to test HDR.
**b** Progeny from HDR cultures was collected at 48 h, grown on fresh cell monolayers and scored for mCherry fluorescence. Example plaques are shown
(scale bar = 300 μm). One hundred plaques per condition were classified as mCh$^+$ or mCh$^-$. **c** Schematic of VACV genomes, gRNA target site and
plasmids used to test HDR at a second site. **d** Progeny from HDR cultures scored for mCherry fluorescence as in (**b**). Graphs depict means and SD of three
independent experiments (**b**) or replicate cultures (**d**) (ns $p > 0.05$ by two-way ANOVA).

A23R-specific gRNAs survived longer after VACV-GFP infection than untreated cells[33]. Targeting of non-essential genes also leads to reduced replication of adenovirus and HSV-1, but comparisons with the effect of targeting essential genes have not been made for these viruses[7]. Finally, we examined virus replication when a template for HDR was introduced in addition to Cas9/gRNA targeting. For the two sites in the genome examined in Fig. 2, cell damage and virus growth were inhibited by Cas9/gRNA complexes (Fig. 3e–h). Taken all together, these data demonstrate that CRISPR/Cas9 is active in the context of VACV infection, presumably reducing replication by cutting viral genomes. We speculate that therefore that the inefficiency in Cas9-mediated genome editing for VACV is due to poor repair, rather than an inability of Cas9/gRNAs to access and cut VACV genomes.

**Cas9 targeting of the VACV genome occurs prior to replication.** Next, we wanted to probe the timing of Cas9 activity against VACV. VACV factories are viral replication centres formed within infected cells. Given that formation of these factories begins early after infection, we postulated that they would only be altered if Cas9/gRNAs acted soon after viral entry. Immunofluorescence was used to visualise factory development in cells transfected with Cas9/gRNA complexes and infected with VACV-mCh. Factories were found in all cultures from 4 hpi, but the number of cells with factories and the size of these factories relative to the nucleus were reduced at 6 and 8 hpi in cultures with Cas9/mCh-gRNA (Fig. 4a, b). These results provide evidence that Cas9/gRNA is acting on VACV genomes very early during infection.

To demonstrate more directly that VACV genomes can be targeted by Cas9 and that this occurs soon after virus entry, we developed a qPCR strategy using primers spanning Cas9-targeted and non-targeted sites. This allowed us to measure genome cleavage at a particular site as a loss of PCR templates relative to the number of viral genomes. VACV-mCh-infected cells were transfected with Cas9 and mCh-, A23R- or non-targeting-gRNAs, and DNA was extracted at various time-points. Using primers

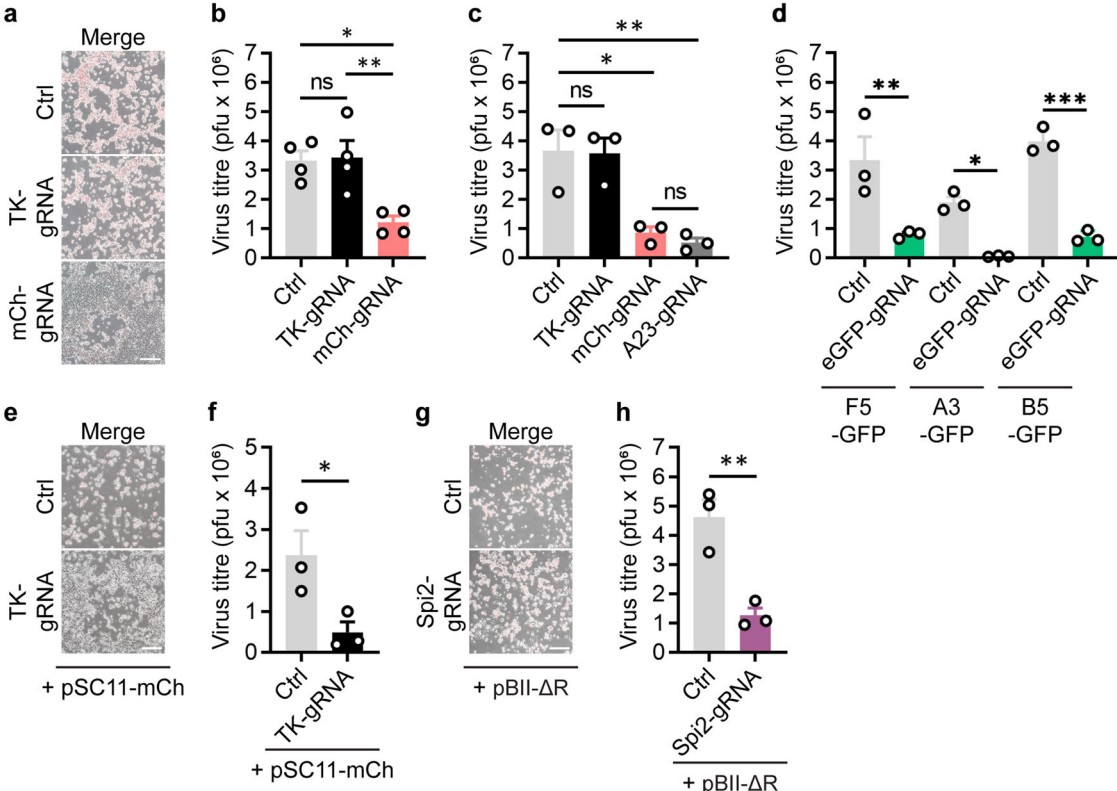

**Fig. 3 Cas9 reduces VACV titre irrespective of cleavage site or presence of HDR template. a** Micrographs show examples of cytopathic effect 48 h after VACV-mCh (MOI 0.05) infection and Cas9/gRNA transfections. **b** Amounts of infectious progeny at 48 h from replicates of the experiment in (**a**) were determined by plaque assay (n = 4, ns p > 0.05, *p = 0.0131, **p = 0.0099). **c** The experiment in (**a**) was repeated with the inclusion of cultures transfected with Cas9/A23R-gRNA (n = 3, ns p > 0.05, *p = 0.0115, **p = 0.0059). **d** Experiments similar to (**a**) but using the VACV F5-GFP, A3-GFP or B5-GFP (as shown) and transfected with Cas9/eGFP-gRNA complexes; Ctrl cultures had no Cas9/gRNA (n = 3, *p = 0.0319, **p = 0.0032, ***p = 0.0003). Plaque assays were used to titrate virus from cultures infected with VACV F5-GFP, A3-GFP or B5-GFP and transfected with Cas9/eGFP complexes. **e–h** Micrographs (**e, g**) and virus production (**f, h**) from VACV-infected cultures 48 h after transfection with Cas9/TK-gRNA and pSC11-mCh (**e, f**) or Cas9-Spi2-gRNA and pBII-ΔR (**g, h**). Scale bars = 300 μm. Graphs display means and SEM from independent experiments (**b, c, f**) or replicate cultures (**d, h**). Significance determined by one-way ANOVA (**b, c, d**) or Student's unpaired t-test (**f, h**) (ns p > 0.05, *p = 0.0458, **p = 0.0071).

and probes to detect *mCherry* or *A23R*, we were then able to determine if these loci were lost and/or replicated poorly in cultures when these sites were targeted by Cas9. In cultures where *mCherry* was targeted, copies of this site were not obviously lost, but viral replication of this part of the genome was delayed compared with other sites, as quantified by the *mCherry* and *A23R* qPCRs, respectively (Fig. 4c, d). Likewise targeting of *A23R* led to a delayed increase in copy number of this gene, but not *mCherry*. To directly compare amounts of the targeted gene versus a second site in the genome, we reanalysed the qPCR data from the experiment above to show these relative to each other. This analysis shows that as expected, the two sites of interest maintained a one-to-one ratio throughout the infection in cultures transfected with Cas9/TK-gRNA, which is unable to target the VACV-mCh genome. However, when Cas9/mCh-gRNA was transfected before infection, *mCherry* amounts fell away relative to *A23R* and vice versa when Cas9/A23R-gRNA was transfected (Fig. 4e, f). These differences were statistically significant from 3 hpi, demonstrating that Cas9/gRNA targets VACV genomes very early after infection.

Taken together, the reduction in number and size of virus factories and loss of uniform genome replication from the earliest times reinforce the idea that Cas9/gRNAs targets incoming VACV genomes. The most likely mechanism of Cas9 action is the cutting of these genomes and separation of the two fragments such that full genome replication cannot be completed and factory development

is inhibited. Dissociation of the cut ends of the genome prior to incorporation in a viral replication complex would also explain the failure of efficient repair by HDR as seen in Fig. 2.

**Cas9 targeting applies selective pressure to VACVs.** The two existing hurdles associated with traditional VACV genome engineering are (1) producing initial recombinants and (2) selecting these from a large background of parent virus. Our starting premise was that CRISPR/Cas9 would improve the production of initial recombinants by facilitating recombination, as in other applications, but our results suggest that Cas9 simply inhibits replication of targeted VACV genomes. Putting these together, we proposed that the best use of Cas9 in the process of engineering VACV might be to select, rather than generate, recombinants. The only requirement for such an application would be that the desired recombinant differs from its parent by the loss of a gRNA target site. To test this proposal, the virus populations shown in Fig. 1, that had a low rate of mCherry⁻ mutants ('initial' in Fig. 5a), were used to infect cells transfected with Cas9/mCh-gRNA. We also included two negative control gRNAs: TK-gRNA that has no target site on the genome and A23R-gRNA that targets the genome at an irrelevant site. In these cultures, 'selected' in Fig. 5a, we saw a distinct pattern where Cas9 targeting of *mCherry* did not appear to protect cells from cytopathic effect as well as in the initial culture, despite markedly reducing mCherry expression,

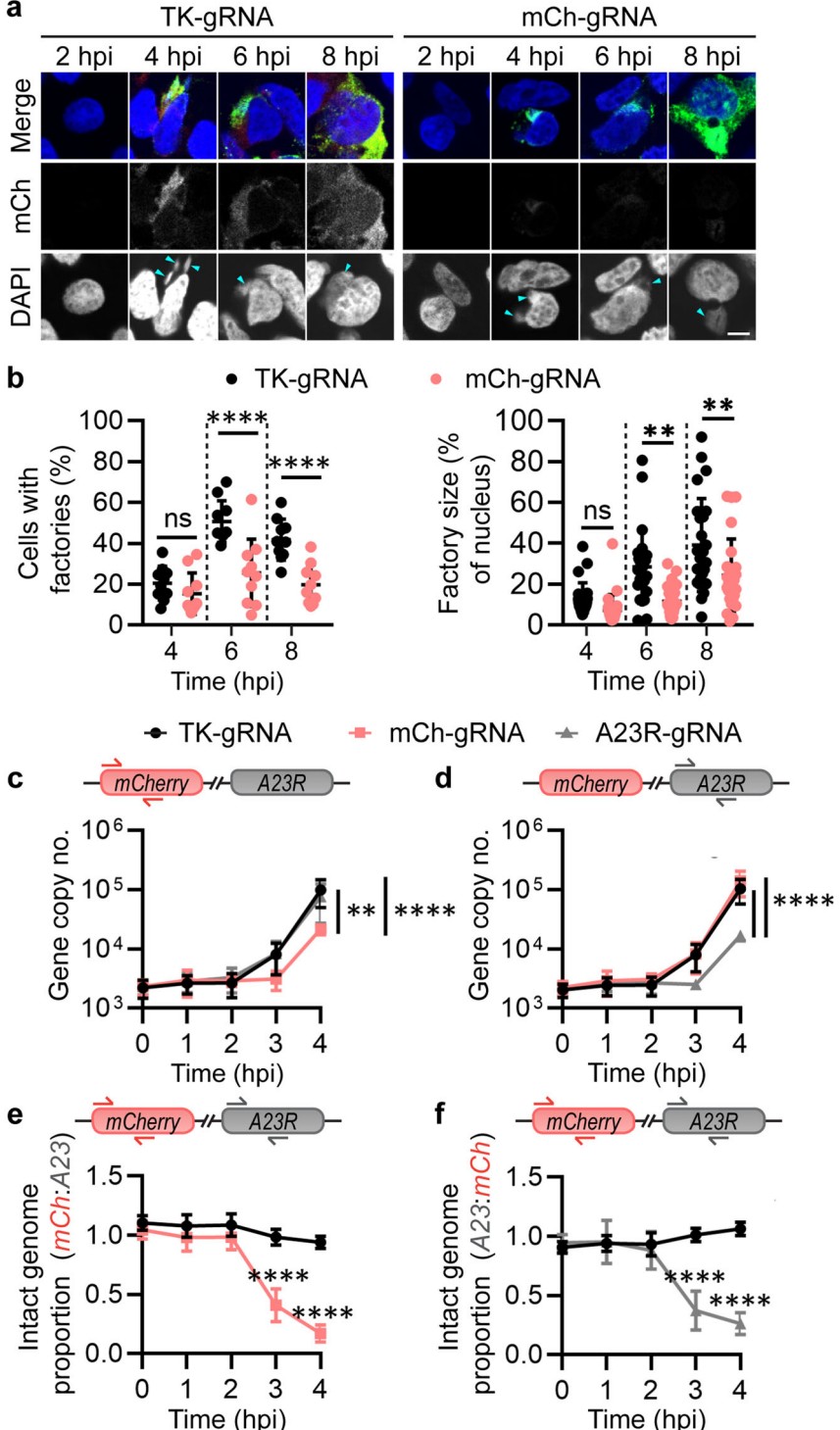

**Fig. 4 VACV genome targeting occurs prior to viral factory formation and replication. a**, **b** Cultures were transfected overnight with Cas9/gRNAs and infected with VACV-mCh (MOI 0.5) for analysis of viral factory formation at the times shown. **a** Immunofluorescence images confirmed infection via anti-VACV primary and anti-rabbit FITC staining (green). Nuclei and virus factories (arrows) visualised with DAPI (blue). Images are representative of 3 independent experiments (scale bar = 5 μm). **b** Number of cells with factories (10 fields of view) and virus factory size relative to the nucleus (25 factories) were measured. Results are shown as means and SD (ns $p > 0.05$, **$p = 0.0013$ or $0.0073$, ****$p < 0.0001$ by two-way ANOVA). **c**–**f** Cells were transfected overnight with Cas9/gRNAs then infected with VACV-mCh (MOI 1, 10 min) for qPCR analysis of genomes at the times shown after infection. Primers used for each qPCR assay are shown above each graph: *mCherry* (**c**) or *A23R* (**d**) for gene copy number measurements, or both (**e**, **f**) for analysis of ratios (means and SD shown; **$p = 0.0023$, ****$p < 0.0001$ by two-way ANOVA, $n = 3$).

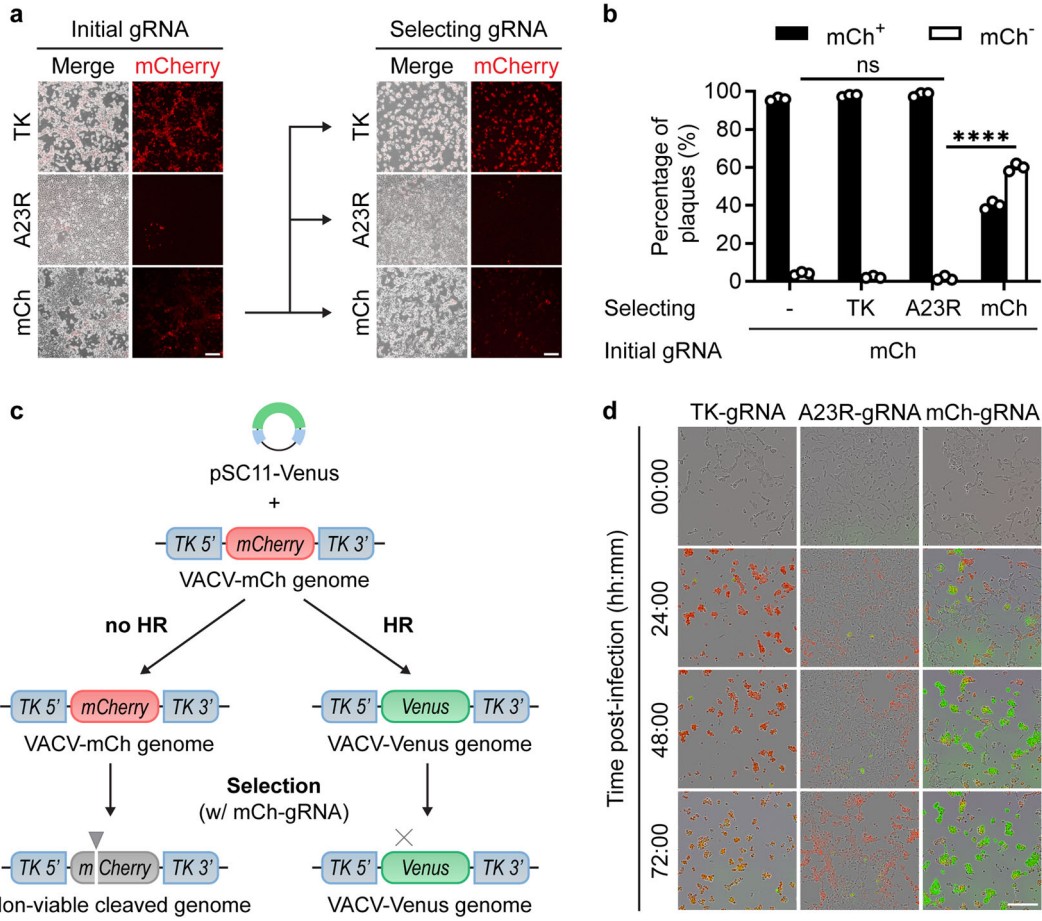

**Fig. 5 CRISPR/Cas9 can be used to select VACV mutants and recombinants. a** Fluorescence micrographs of VACV-mCh-infected, Cas9/gRNA-transfected cell cultures, and secondary cultures infected with initial cultures and selected with Cas9/gRNAs. Scale bar = 100 μm. **b** Progeny from cultures in (**a**) were grown on fresh cell monolayers and 100 plaques for each were scored for fluorescence (SD obscured by data points; ns $p > 0.05$, ****$p < 0.0001$ by two-way ANOVA, $n = 4$). **c** Model for studying selection after homologous recombination (HR) showing viral genomes, repair plasmid (pSC11-Venus) and Cas9/mCh-gRNA used for selection. **d** Images taken from live-cell imaging of the selection cultures described in (**c**), with control gRNAs included: TK-gRNA does not target the genome, A23R-gRNA targets the genome at an irrelevant locus. Images are representative of 3 independent experiments. Scale bar = 200 μm.

hinting at the presence of a substantial mCh⁻ population (Fig. 5a). By contrast, the use of Cas9/A23R-gRNA, considerably reduced cytopathic effect in both the 'initial' and 'selected' cultures. This was confirmed when plaques grown from the selection cultures were scored for fluorescence, revealing that mCh⁻ plaques had increased to 60% when selected by the mCh-gRNA, but remained very low when the TK-gRNA or A23R-gRNA were used (Fig. 5b). Total virus output was also greater from the culture where mCh-gRNA was used for selection, likely reflecting outgrowth of the mCh⁻ population during selection (Supplementary Fig. 3). To demonstrate selection visually, we used traditional homologous recombination to achieve the first step in replacing the *mCherry* gene in VACV-mCh with *Venus*. The progeny of this recombination culture were then selected by a round of replication in cells that were transfected with Cas9/mCh-gRNA or control RNPs (Fig. 5c). Live-cell imaging of the selection cultures allowed us to visualise selection of Venus⁺ VACV in real-time (Fig. 5d, Supplementary Video 1). In cultures transfected with Cas9/mCh-gRNAs, the outgrowth of Venus⁺ virus was evident after 24 h and continued over the next 2 days. By contrast, when the non-targeting Cas9/TK-gRNA control was used, any Venus⁺ virus was swamped by the mCherry⁺ VACV-mCh parent at all times. Finally, Cas9/A23R-gRNA that targets an irrelevant site in the genome inhibited all virus growth as indicated by dramatically

reduced cytopathic effect and fluorescence of either colour. Overall, these results demonstrate that Cas9/gRNA complexes can be used as a robust and specific selection tool for rapidly eliminating parent virus from a population, allowing expansion of desired recombinants.

**Optimised method for generating recombinant VACVs by Cas9 selection.** The ability to rapidly select recombinant VACVs has wide implications for vaccine vector and oncolytic virus development. We thus attempted to develop a streamlined selection pipeline for scientific and therapeutic applications. Our method has three steps, initial recombination then two rounds of Cas9 selection, taking a total of 6 days (Fig. 6a). For initial validation, we used the same fluorescent virus model as in the last figure, replacing *mCherry* in VACV-mCh with *Venus*. This allowed the progress of selection to be checked by keeping some virus aside at each step and scoring plaques for fluorescence. After initial recombination, Venus was detected in 5% of the progeny plaques, with some of these being double-positive (DP) for Venus and mCherry (Fig. 6b). The first round of selection with a Cas9/mCh-gRNA reduced the total yield of virus but more importantly increased the fraction of Venus⁺ plaques dramatically to 80%, the majority of which were Venus⁺ only (Fig. 6b, Supplementary Fig. 4a). One concern with this

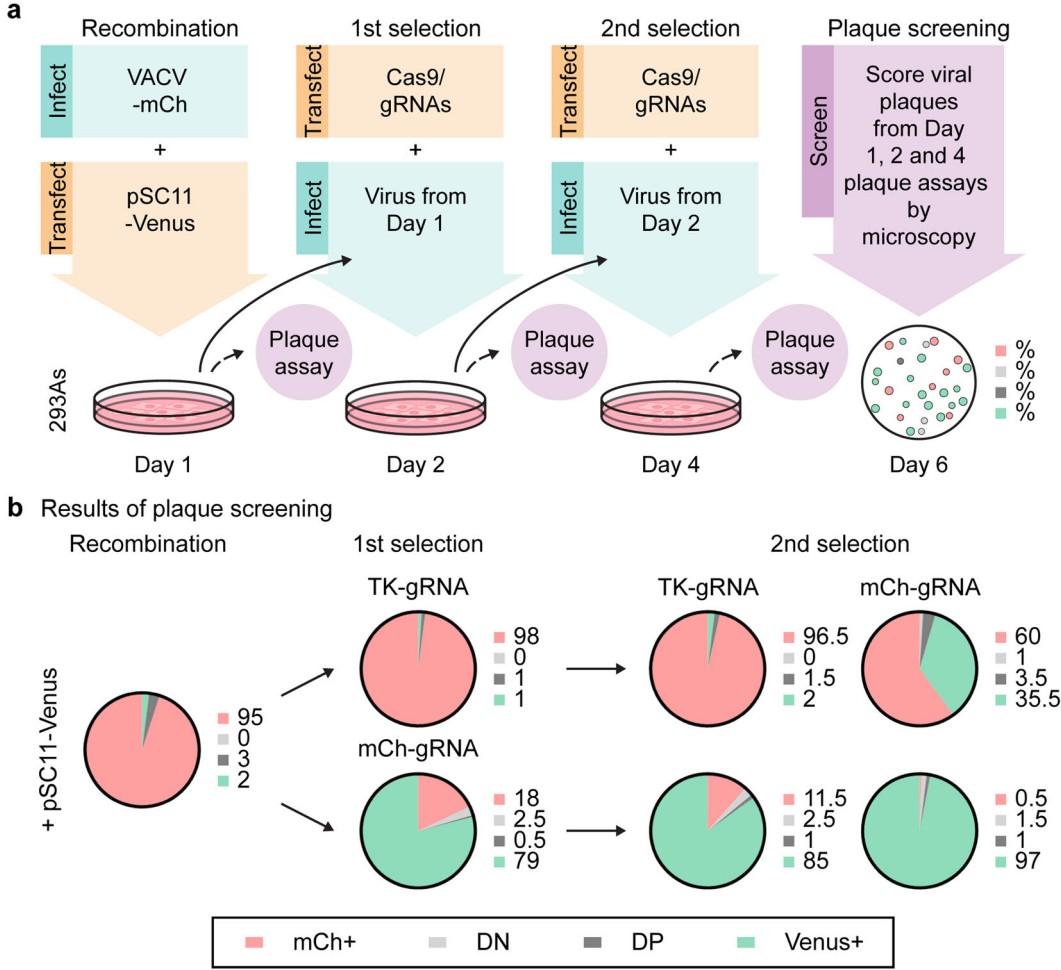

**Fig. 6 Rapid creation of a recombinant VACV using Cas9 selection. a** Schematic showing a rapid 3-step protocol to exchange *mCherry* in VACV-mCh for *venus*, including the steps taken to track selection. **b** Proportions of mCherry-positive (mCh[+]), double-negative (DN), double-positive (DP) and Venus-positive viruses (Venus[+]) following homologous recombination and Cas9 selection are shown. One hundred plaques were counted per condition. Pie charts and adjacent numbers are means across two independent experiments.

method is the possibility of editing by NHEJ and outgrowth of these unwanted viruses during the selection steps, which was seen when there is no recombinant virus to be selected (Supplementary Fig. 4b). However, in our main experiment where we were selecting Venus[+] virus from a background of mCh[+] parent, there was a very low frequency of editing by NHEJ as reflected by plaques with neither fluorescent marker (double-negative) after this first selection step (Fig. 6b). The second selection step then almost completely removed the remaining parental virus, and total virus output was now similar to a non-targeted culture (Fig. 6b, Supplementary Fig. 4a). Importantly, there was no significant increase in plaques with no fluorescence, again pointing to very low levels of editing by NHEJ such that these viruses are unable to compete with the desired recombinant VACV during selection and are not a problem. We also tested A23R-gRNA, which cuts VACV genomes at an irrelevant site relative to our desired recombinant. While this reduced virus titre, it did not lead to selection of Venus[+] plaques, which were lost completely after the second selection step (Supplementary Fig. 4c, d). These data suggest virus selection by this method is gRNA-specific, as expected, and not a side-effect of the reduction of virus growth caused by targeting the genome at other sites. This method generates high frequencies of desired recombinant VACVs in less than a week such that only a handful of plaques would need to be screened to select recombinants for the establishment of clonal stocks.

**CRISPR/Cas9 selection can be efficiently used to select marker-free viruses.** Our previous example used fluorescent viruses, but a better test of the method would be to rapidly generate a recombinant VACV with a small edit and no observable phenotype in vitro, something that can often take weeks to months. Marker-free viruses are considered better research tools and marker genes are typically unacceptable in therapeutics and vaccines. VACV Spi2 protein, encoded by *B13R*, is a serine protease inhibitor that influences viral pathogenesis but not growth or plaque characteristics in vitro and was chosen as a target due to ongoing projects within the laboratory[42–50]. To construct Spi2-knockout viruses, we designed a recombination vector to replace 20 bp of *B13R* with a sequence that introduces a premature stop codon, removes the recognition site for a Spi2-specific gRNA and adds two restriction enzyme sites (Fig. 7a). Starting with wild-type VACV, we used this plasmid and Cas9/Spi2-gRNA in the 3-step protocol described in the last experiment, except in this case 10 plaques grown from the progeny of each culture were tested by PCR for evidence of the desired recombinant (Fig. 7b). To ensure we were not seeing a non-specific effect of Cas9, we again used a non-targeting gRNA (mCh-gRNA) as a control. Compiling data from two independent experiments, we detected Spi2 recombinant viruses by PCR at a frequency of 55% after two rounds of Cas9/Spi2-gRNA selection (Fig. 7c). Again, selection of desired recombinants was accompanied by reduction in total virus

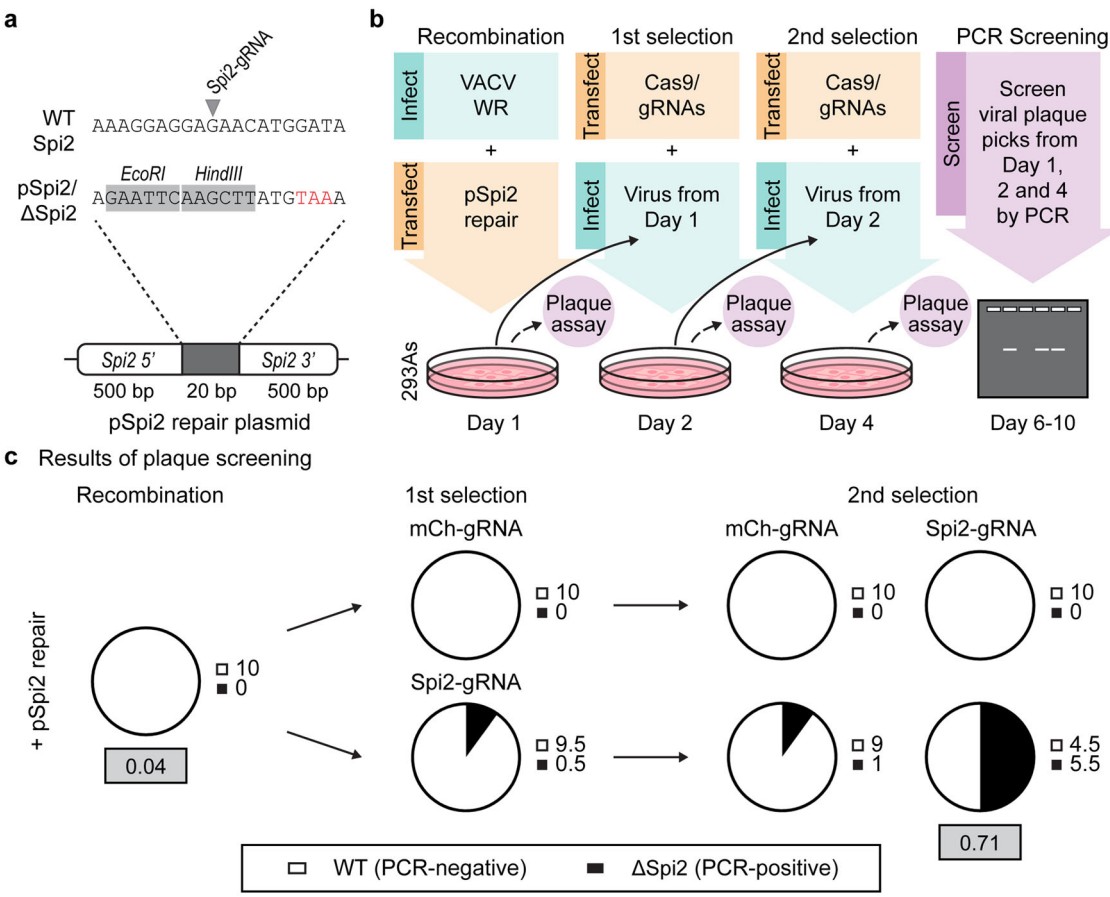

**Fig. 7 Rapid creation of a marker-free VACV. a** Design of a 20 bp section used to knock out VACV Spi2 showing premature stop (red), restriction sites (grey) and Spi2-gRNA site on parent genome. **b** Rapid protocol to create a VACV Spi2 knockout, showing steps taken to track selection. **c** Proportions of parent and desired recombinant after each step as determined by PCR. Pies and adjacent numbers show the average from 10 plaques at each step from two independent experiments. Grey boxes show frequency of the desired recombinant after analysing 100 plaques after recombination and after two rounds of selection.

production, implying suppression of parent virus replication (Supplementary Fig. 4a). We also confirmed the identity of the Spi2 recombinant viruses by restriction digest and DNA sequencing (Supplementary Fig. 4b). Of the 20 plaques from the two experiments, all had either parent or desired recombinant sequence, as expected from the initial PCR screen and confirmed by restriction digests. Notably, none of these sequences showed evidence of editing by NHEJ, again suggesting that this process is not a confounder in our selection method. To gain a better estimate of the efficiency of Cas9/Spi2-gRNA selection, we screened an additional 100 plaques grown from the progeny of the recombination and second selection cultures. Using this method, we found that Spi2-knockout frequency in the population increased from 4 to 71% after Cas9-mediated selection (Fig. 6c, grey boxes). These data demonstrate that our method using Cas9 selection allows reliable generation and initial plaque isolation of a marker-free VACV in less than one week. Further, the frequency of desired recombinants after two selection steps is high enough that it would be feasible to identify these isolates directly by DNA sequencing.

## Discussion
CRISPR/Cas9 has revolutionised genome engineering and, as a large dsDNA virus, VACV is an obvious candidate for use of this tool. Indeed, there have been descriptions of such an application to VACV, the first being published five years ago, even if the approach has not become widespread in the field[32,33]. These authors

concluded that CRISPR/Cas9 facilitated engineering by NHEJ and increasing recombination rates, so it is necessary to reconcile our findings here with these prior reports. Our results agree that NHEJ can occur after Cas9 targeting of VACV genomes, but the low frequency with which this occurs was not previously reported. Further, the previous study examining NHEJ used low multiplicities of virus and cultures were maintained over many days[33]. This means that Cas9 can be acting both to initiate NHEJ at a low frequency and then more potently to select viruses with alterations at the target site as they spread through the one culture. Indeed, Luteijin et al. note this possibility in the context of the escape of VACVs from Cas9/gRNA control where mutations caused by NHEJ do not compromise virus replication[33]. However, they did not take the next step to suggest that this selection might be driving the apparent success of their engineering efforts more generally. This brings us to whether Cas9 targeting increases rates of homologous recombination. Again in the published reports, the low initial virus inocula and length of time of culture means that it was not possible for the authors to distinguish between increased rates of recombination and recombination followed by selection. Given that all other applications of CRISPR/Cas9 discuss cutting of genomes and repair as if they are linked events, where recombinant VACVs were able to be made more easily in the presence of Cas9/gRNAs it was logical to assume that the mechanism at play here was the same. We have only come to a different conclusion due to a deliberate dissection of these two parts of the process. The motivation for this was several years of failure and direct comparisons

with results from our development of CRISPR/Cas9 editing of HSV, where it was easy to demonstrate improvements in recombination frequency[6]. We cannot rule out the possibility that Cas9 targeting achieves an improvement in recombination frequency for VACV using some cell types or protocols. However, we have never found this to be a robust effect in over 5 years of consistent work in multiple cell types and doing our best to repeat published protocols; Fig. 2 here is representative of our experience, both with RNPs and plasmid-delivered Cas9/gRNA. Further, the experiments shown here include gRNAs targeting five different genes spread across the VACV genome in six different locations (Supplementary Fig. 1), which strongly suggests that our data are truly representative. Finally, we note that the speed of selection differs across our experiments (e.g. compare Fig. 6 and Fig. 7), which may be linked to the efficiencies of the different gRNAs in directing Cas9.

Cas9, in its native form in *S. pyogenes* and as used here lacks a nuclear localisation signal[51]. So superficially it is perhaps unsurprising that it was able to target viral genomes in the cytoplasm. However, it is not known to what extent poxviruses genomes are exposed in the cytoplasm during an infection cycle, much of which is contained in dense, membrane-bound factories that might be an impediment to Cas9 access[19,20]. Nevertheless, Cas9 targeting appears to be highly efficient (Fig. 4)[31–33]. Access to factories by non-viral proteins has been seen previously[52], but our results (Fig. 4) suggest that Cas9 acts very early, between the release of the genome from incoming virions and the full establishment of viral factories. This matches the outcome of efficient targeting, which was a global decline in viral gene expression, replication and cytopathic effect, which most likely occurs due to disruption of uniform replication across the genome and factory development (Figs. 3 and 4). Further, factories and viral replication are requisites for VACV-mediated recombination and the inhibition of these by genome targeting prior to full factory establishment may explain why HDR is not efficient for VACV genomes targeted by Cas9. Alternatively, if genomes are cut prior to factory formation, the resulting segments may disperse and this separation would also preclude HDR (Fig. 3). Whatever the mechanism, the early targeting of poxvirus genomes would seem to underlie the efficient targeting by Cas9 on one hand, yet the inefficient HDR on the other.

NHEJ is a predominantly nuclear process. However, VACV is defined by its nuclear independence, even possessing the ability to replicate within enucleated cells[53,54]. This might suggest that cleaved VACV genomes may not have access to known mediators of NHEJ repair, such as DNA ligase IV[9,10,55]. However, others have already shown that NHEJ is not precluded for VACV and that DNA ligase IV is a critical factor, despite the cytoplasmic location[33]. DNA ligase IV is typically recruited by Ku to sites of DSBs alongside XRCC4, a driver of nuclear localisation[56–60]. We note that while Ku has been found within viral factories, presumably in its role in pathogen DNA sensing, DNA ligase IV and XRCC4 were not found[33]. The relative inefficiency of NHEJ that we show might be due to the paucity of these factors in the cytoplasm, or that ligase IV is not primed for the efficient repair of cytoplasmic genomes (Fig. 1 and Supplementary Fig. 4). Indeed, overexpression of ligase IV can increase NHEJ rates for VACV[33]. Efficiency rates with endogenous ligase IV levels have yet to be reported, but our data would suggest it lies between 5 and 10%, much lower than the supposed rate of genome cleavage (Figs. 1, 3 and 5). Importantly for the development of Cas9 as a selection tool for VACV, the efficiency of NHEJ is low enough that it does not drive an accumulation of mutated parent genomes instead of desired recombinants. These would have appeared as non-fluorescent plaques in our experiment shown in Fig. 6 and while such plaques were found, they appeared at low frequency after the first selection step (2.5%) and fell after the second round of selection (1.5%). In the marker-free virus experiment in Fig. 7, none of the 20 plaque isolates

sequenced had changes consistent with repair by NHEJ, all matched either the desired recombinant or the parent (Supplementary Fig. 5). None of this is to say that NHEJ could not be used to knock out VACV genes, but we would recommend using it in conjunction with additional rounds of selection with the same Cas9/gRNA. The potential of such a method can be seen in our experiment in Fig. 5, where a single round of selection achieved an enrichment of mutant virus such that it was more frequent than the parent.

In conclusion, we have turned the inefficiency of DNA repair after Cas9-mediated VACV genome targeting to our advantage and in doing so found an alternative use for CRISPR/Cas9: the selection of recombinants, or other genetic variants. There are other examples where repair mechanisms after Cas9 cutting are inefficient, most notably in some bacterial species, and in these cases a pivot to using Cas9 as a selection agent, rather than the initiator of mutation or recombination, may find similar utility. In the case of VACV, well-understood but inefficient conventional methods for engineering VACV can be super-charged by a strong and rapid selection that requires no marker genes and can be applied to any site in the genome. The three-step method we demonstrate is reliable and rapid as long as efficient transfection of Cas9/gRNA can be achieved. If a single site of the genome is to be targeted regularly, this requirement for transfection could be avoided by creating a stable cell line expressing Cas9 and the required gRNA, greatly simplifying the method. The speed of making recombinant VACVs by this method is noteworthy because it extends the viability of this viral platform. Applications that are now within easy reach are personalised medicine where therapeutics need to be made quickly on a person-to-person basis and vaccines for emerging disease outbreaks such as COVID-19, where turnaround time is paramount.

## Methods

**Cell lines and viruses**. Cell lines used in this study include 293A (human embryonic kidney cells) and BS-C-1 (African green monkey kidney epithelial cells). Both lines were maintained in Dulbecco's modified Eagle medium, high glucose (DMEM, Gibco) supplemented with 10% (v/v) heat-inactivated foetal bovine serum (FBS) and 2 mM L-glutamine (Gibco) (D10). Cells were grown at 37 °C in 5% $CO_2$ for the duration of experimentation.

A number of VACVs were used within this study. Non-recombinant, VACV Western Reserve strain (VACV WR), was a kind gift of B. Moss, NIH. A recombinant virus possessing *lacZ* and *mCherry* gene insertions within the thymidine kinase (*TK*) gene was denoted VACV-mCh. This virus was generated by homologous recombination using the pSC11-mCh plasmid described below, and screened for fluorescence prior to plaque purification. VACV F5-GFP and VACV-A3-GFP have been described previously, and express eGFP fusions at the carboxy-terminus of F5 and amino-terminus of A3, respectively[37,38]. VACV B5-GFP has the same eGFP construct as A3, but fused to the 3′ end of gene *B5R* in its native location. This virus has not been described elsewhere but was made by conventional homologous recombination with a transfer plasmid by published methods[61]. The identity of VACV B5-GFP was verified by PCR fragment length and sequencing of the region of the genome that was manipulated. VACV infections were carried out as follows. Briefly, cells were exposed to virus diluted in serum-free DMEM supplemented with 2 mM L-glutamine (D0) for 1 h before rescuing with 2% (v/v) FBS and 2 mM (v/v) L-glutamine-containing DMEM (D2). Virus titres were quantified by standard plaque assay on BS-C-1 cells, whereby confluent cell monolayers were infected with 10-fold serially diluted virus in D0 for 1 h, before media removal and overlay with 0.4% (w/v) carboxymethyl cellulose in D2. At 48–72 h post-infection (hpi), plaques were either visualised microscopically or by crystal violet staining using a 0.1% (w/v) crystal violet and 15% (v/v) ethanol fixing solution.

**Plasmids and single-guide RNAs (gRNAs)**. Four homologous recombination plasmids were used for the majority of the work in this study. Briefly, pSC11-mCh contained an *mCherry* gene (NCBI #MK160997.1) under the control of a p7.5 early VACV promoter, which has been inserted into a previously characterised pSC11 VACV recombination vector[62,63]. This backbone possesses a *lacZ* gene under the control of a p11 late VACV promoter and regions of homology to the *TK* gene of VACV[64]. The pSC11-Venus plasmid was created from the same backbone, but instead possesses a *venus* gene (NCBI #DQ092360.1) under the p7.5 promoter. pBII-ΔR was used previously to generate a recombinant VACV with a 23 kb deletion in the right arm of the genome[39]. In this case, recombination was utilised to replace VACV WR genes from *Spi2* to *C11R* with mCherry. Alternatively, the pSpi2 plasmid was synthesised by Integrated DNA Technologies, and contains a modified *Spi2* gene insertion in a pUCIDT-Amp commercial vector backbone.

The gene insert has 500 bp homology arms which flank a region of interest in Spi2. This 20 bp sequence contains a premature stop codon at position 147, introduced *EcoRI* and *HindIII* restriction sites and bases that have been altered for evasion of recognition by the Spi2-gRNA (5′-AAAGGAGGAGAACATGGATA-3′).

Other gRNA sequences used in this study include those specific to *TK* (TK-gRNA: 5′-TCACAGAATTCAACAATGTC-3′), *mCherry* (mCh-gRNA: 5′-GGAT AACATGGCCATCATCA-3′), *eGFP* (eGFP-gRNA: 5′-GCTGAAGCACTGCACG CCGT-3′)[33] and *A23R* (A23R-gRNA: 5′-GAAAGAACGCATTTCCTCAG-3′)[33]. All gRNA sequences were ordered as single RNA molecules, where the gene-specific sequence prefaced a common tracrRNA sequence (5′-GTTTTAGAGCT AGAAATAGCAAGTTAAAATAAGGCTAGTCCGTTATCAACTTGAAAAA GTGGCACCGAGTCGGTGC-3′). A list of each virus used in this study can be found in Supplementary Fig. 1.

**CRISPR/Cas9 transfections**. The standard method for the application of CRISPR/Cas9 in this study followed an infection/transfection approach. Briefly, a 1 h VACV infection (MOI 0.05) preceded transfection of sub-confluent 293A cells with ribonucleoprotein (RNP) complexes and in some cases templates for homologous recombination. These pre-formed RNP complexes contained Cas9 nuclease (20 μM) lacking a nuclear localisation sequence (NLS) (New England Biolabs) and 2 μg of appropriate gRNAs. Complex formation was allowed to progress for 30 min prior to transfection. All transfections were carried out with Lipofectamine 2000 (Life Technologies) in D0 media for 4 h, before replacement with fresh D2 media and incubation for 48 h. At this point, wells were imaged to assess fluorescence and cells and supernatant were retained for further analysis. Virus titres and CRISPR-editing capacity were monitored using flow cytometry. Briefly, scraped cells were washed in PBS, fixed in 1% (v/v) paraformaldehyde (PFA) and resuspended in 2% (v/v) FBS-containing PBS. Venus and mCherry expression, as well as cell counts were captured with an ACCURI (BD Biosciences) cytometer. Cytometry data was compiled using FlowJo software (v10.6.2). In addition, plaque assays (described above) were used to titrate viruses on BS-C-1 cells and count edited viral plaques. Note, for experiments involving recombination, plasmids were first linearised by overnight digestion and then co-transfected with the Cas9/gRNA complexes.

To investigate the temporal effects of transfection and infection on CRISPR/Cas9-mediated gene editing, this protocol was reversed. That is, cells were first transfected for 4 h before reagents were removed and the appropriate virus was added to cells. Again, transfection/infections were allowed to progress for 48 h.

**Cas9 selection**. For Cas9 selection experiments, sub-confluent 293A cells were first infected with VACV (MOI 0.05) for 1 h, then transfected with linearised plasmid alone for 24 h. Cells and supernatant were obtained as before and used to infect fresh 293A cells that had been pre-transfected with Cas9/gRNA complexes. At 48 h, this process was repeated to give a total of two rounds of CRISPR/Cas9 selection and analysed by fluorescence or PCR screening techniques. Note that at each point, virus titres were determined by plaque assay.

**Viral genome quantitation by qPCR**. To assess genome copy numbers and relative proportions of whole and cleaved genomes, two qPCR analyses were performed using TaqMan probe chemistries. To enhance the sensitivity of these assays, the CRISPR/Cas9 transfection/infection protocol above was followed with minor changes. First, 293A cells were transfected for 4 h, as before, with pre-formed Cas9/gRNA complexes. Subsequently, infections were carried out with a higher titre of VACV-mCh virus (MOI 1) for a shorter period of time (10 min). At this point, cells were washed in PBS to remove unattached virions and re-supplied with fresh D2 media. Infections were then allowed to progress for the specified times prior to DNA extraction.

For DNA isolation, cells and supernatant were collected and snap-frozen in a dry ice and ethanol bath. The QIAamp DNA Mini Kit (Qiagen) protocol was followed to retrieve DNA from each sample, the concentration of which was determined by Nanodrop (ThermoFisher) spectrophotometer analysis. For DNA standards, DNA was isolated from a sucrose cushion purified stock of VACV WR. A Qubit dsDNA BR (Broad-Range) Assay Kit and Qubit Spectrophotometer (ThermoFisher) were utilised to determine the concentration of the DNA standard.

For qPCR analysis, custom TaqMan Gene Expression assays were designed by ThermoFisher to analyse *A23R* and *mCherry* gene levels. Proprietary probes for these genes were complexed to FAM and VIC dyes, respectively, to facilitate multiplexing. With these pre-designed assays, 1.25 μg DNA was added to each well alongside 5 μL of TaqMan Fast Advanced Master Mix (ThermoFisher) and 0.5 μL of the probe and primer pre-mix. All qPCRs were performed using the QuantStudio 1 (Applied Biosystems) or 7900HT Fast Real-time PCR (Applied Biosystems) apparatuses. The thermocycling conditions involved a 2 min, 50 °C and 10 min, 95 °C holding stage, followed by 40 cycles of 95 and 60 °C for 15 s and 1 min, respectively. Gene copy number was calculated relative to the DNA standard, whereby 10-fold dilutions of $10^9$ VACV-mCh genomes were used to generate standard curves for *A23R* and *mCherry*. Note that calculations for the DNA concentration to genome copy number conversion were done using a known VACV WR sequence (NCBI #NC_006998).

**Immunofluorescence analysis**. To examine viral factory formation, immunofluorescence assays were performed. 293A cells were seeded on glass coverslips that had been pre-coated with Poly-L-lysine (Sigma-Aldrich) according to the manufacturer's instructions. At 70–80% confluency, cells were transfected with the appropriate Cas9-gRNA complexes, as described above. Four hours post-transfection, media was replaced with D2 for 16 h, before infecting cells with VACV-mCh (MOI 0.5) in D2. Infections were allowed to progress for the specified times before fixation of cells in 1% (v/v) PFA. To confirm the presence of infection, fixed cells were blocked in 1% (w/v) BSA and 2% (v/v) FBS in PBS for 20 min. Samples were then probed with a rabbit anti-VACV antibody and a corresponding secondary, anti-rabbit IgG-FITC (Silenus, 1:200)[37]. Nucleic acids were visualised with Hoechst 33342 (Invitrogen, 1:10,000). Coverslips were mounted onto glass slides with a mounting media containing 10% (w/v) Mowiol (Poly(vinyl alcohol)) 4–88, 25% (w/v) glycerol and 100 mM Tris (pH 8.5) and allowed to dry overnight prior to examination. All fixed cell imaging was conducted on the Zeiss Axio Observer wide-field microscope and the Leica SP5 confocal system.

**Live-cell imaging**. For live-cell imaging, the IncuCyte Live Cell S3 system (Sartorius) was used. Briefly, the CRISPR/Cas9 selection protocol was followed as described above with cells at 70–80% confluency. All imaging was then conducted using the 6-well plate format, with micrographs taken at 20 min intervals from 1 to 72 hpi. During this time cells were incubated at 37 °C and 5% $CO_2$. ImageJ software (v1.53) was used to annotate video files.

**Plaque isolation of virus and PCR screening**. To screen viral plaques in our marker-free model of CRISPR selection, we used a standard PCR screening method. Briefly, following Cas9 selection, viruses were grown on BS-C-1 cells, single plaques were identified by microscopy and cells and virus were collected with a pipette and added to D2 media for expansion before testing by PCR. This involved freeze-thawing three times, before plating these virus isolates onto uninfected BS-C-1 cells in a 96-well plate and allowing viral replication to progress for 3 days. At this point, cells were lysed by addition of proteinase K (10 μg/mL) diluted in 1X ThermoPol buffer (New England Biolabs) and PBS. Following a freeze-thaw step at −80 °C, lysates were heated at 56 °C for 20 min, then 85 °C for 10 min. DNA obtained in this manner was used as template for a PCR with insertion specific Spi2 screening primers (Spi2 screening FW: 5′-GAAGAATTCAAGCTTAGTGAAAAGG-3′, Spi2 REV: 5′-CTTACATCTACCATTTCCGTCG-3′) and the presence of bands was evaluated by agarose gel electrophoresis. Spi2 PCR products were also digested overnight with *HindIII* to confirm the presence of the inserted 20 bp by RFLP analysis on agarose gels. Note for this experiment, PCR primers used were a FW primer specific to both wild-type and recombinant Spi2 (Spi2 sequencing FW: 5′-GTATTCATTTCTCCA GCGTCA-3′) with the Spi2 REV used above.

**Sanger sequencing**. To sequence the Spi2 region of the VACV genome, a DNA fragment was amplified by standard PCR using the Spi2 sequencing FW and Spi2 REV primers described above. The amplified fragment was separated from the primers using agarose gel electrophoresis and subsequently purified by gel extraction according to the manufacturer's protocol (Machery-Nagel Nucleospin Gel and PCR Clean Up Kit). DNA yield and purity was assessed by Nanodrop spectrophotometry (ThermoFisher). For sequencing, a 20 μL reaction containing 40 ng DNA, 1 μL BigDye Terminator (ThermoFisher), 3.2 pmol Spi2 FW sequencing primer and 3.5 μL 5X sequencing buffer. Samples were cycle sequenced at 94 °C for 5 min, followed by 10 cycles of 96 °C (10 s), 50 °C (5 s) and 60 °C (4 min), then a 4 °C hold stage. Prior to Sanger sequencing using the AB3730 capillary sequencer, DNA was ethanol precipitated and air-dried. All sequencing was then performed by the Australian Cancer Research Foundation Biomolecular Resource Facility (BRF) at the Australian National University.

**Statistics and reproducibility**. Sample sizes were determined based on the assay. For virus plaque analysis, 30–50 plaques are considered standard for analysis. We increased this number to 100 to gain a better insight into virus frequencies within mixed populations. For flow cytometry analysis, 25,000 cells were considered sufficient per sample. All other samples were analysed in their entirety. All experiments were replicated 2–3 times independently.

All statistical analyses were performed using GraphPad Prism 8 (v8.4.2) software. Typically, one- or two-way ANOVA were used with a Tukey post-test to make multiple pair-wise comparisons, or where appropriate a Student's unpaired *t*-test. Unless stated otherwise error bars refer to SD.

**Reporting summary**. Further information on research design is available in the Nature Research Reporting Summary linked to this article.

## Data availability

The data that support the findings of this study are available in this manuscript or from the corresponding author upon request. Source data for Figs. 1b–e, g, 2b, d, 3a–h, 4a–f, 5a, b, 6b and 7c and Supplementary Figures 2, 3 and 4 and 5a are provided in Supplementary Data 1.

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

## Acknowledgements

We thank B. Moss, NIH for VACV WR, B. Dobson, ANU for VACV F5-GFP and the pBII-ΔR plasmid, Yik Chun Wong, ANU for VACV-A3-GFP and staff of the ACRF Biomolecular Resource Facility at ANU for sequencing services. We thank S.M. Man for live-cell imaging equipment. D.C.T. is funded by grants and fellowships from the NHMRC: APP1104329, APP1084283 and APP1126599 and ARC: DP190101325.

## Author contributions

A.G., S.S. and D.C.T. conceptualised the study. A.G. and S.S. performed the experiments. A.G., S.S. and T.S. generated viruses. A.G., S.S. and D.C.T. conducted analyses and interpreted results. D.C.T. obtained funding. A.G. and D.C.T. wrote the manuscript. All authors reviewed it.

## Competing interests
The authors declare no competing interests.
