## [Peer Review File · Communications Biology]

Reviewers' comments:

Reviewer #1 (Remarks to the Author):

This manuscript deals with using CRISPR/CAS9 system to deselect a population of recombinant virus. Traditionally this system has been used to create a high rate of recombination, which the authors did not see. However, they were able to use this system to efficiently cleave the parental virus, and increase the selection of there recombinant. Selecting for a pure recombinant virus is the most time consuming step in this process. The authors also show the utility to make marker free virus.

The study was well throughout and well implemented. All the controls were used properly.

The discussion was encompassing of the field. I have no changes to this manuscript.

Reviewer #2 (Remarks to the Author):

The manuscript entitled " Rapid virus engineering using CRISPR/Cas9 as a selection tool" reported repurposing Cas9 targeting as a selection tool for enriching recombinant vaccinia virus. I have the following concerns :

Major :

1. Is the selection effect of Cas9 site-specific? According to Figure 3, Cas9 reduces VACV titer regardless of the cleavage site. However, in Figure 4c and d, the copy number of mCherry decreases in response to mCh-gRNA but not A23R-gRNA, or vice versa. I believe the site effect of Cas9 targeting should be one key factor in this selection pipeline and be studied further.
2. The authors showed low efficiency of Cas9-mediated NHEJ repair (less than 10%) in VACV. However, in the selection process, it is possible that NHEJ mutants generated during the first Cas9 targeting could be enriched in the following steps, thus impairing the efficiency of "selection". I believe this could be a challenge especially when the initial recombination rate is low.

Minor:

1. If you want to claim high gene editing efficiency, one site is enough. But if you show the low efficiency, multiple sites should be tested considering that high level of gene editing has been reported earlier.
2. It is better to indicate the name of virus since only VACV was studied.

Reviewer #3 (Remarks to the Author):

Gowripalan A. et al reported development of a method to enrich conventional poxvirus recombinants for making recombinant viruses without the aid of marker genes in less than a week. They utilized the inhibition of viral replication by Cas9 nuclease to eliminate non-recombinant population. The principle of the method is simple but I think there is no improvement or findings in this report. In addition, the use of the method is very limited because the state-of-art of selection is totally dependent on replication inhibition by Cas9, which could be seen only the vaccinia virus. The recombination rate in the infected cell or viral factory is not improved at all and considered to be remained very low.

There are some issues I concern about the experiments in the text.

1. In the video file, not all cells show fluorescence of mCherry. How this could be explained? Did authors determined infection efficiency of VACV? Seeing the video, there looks about 30% of uninfected or without mCherry fluorescent cells exist.

2. I did not find any improvement about recombination efficiency of infected cells. It is true that the authors used "elimination strategy", which is different from the previous reports. But I think there is no novelty in the method. Did authors find any difference or improvement in recombination efficiency in their method?

3. As they record time-lapse change of fluorescence from mCherry to Venus, they are able to observe and count cell numbers those have recombination events. As far as I see, the recombination events are very low and only replicated cell population is growing.

4. In experiments for Figures 6 and 7, they should use the same protocol to select recombinant-positive VACV. The efficiency of the selection is different; they get almost 100% Venus positive in Fig 6, although they only get 50-70% positive in Fig 7. How this can be explained?

Response to Reviewers Comments

Reviewer #1 (Remarks to the Author):

This manuscript deals with using CRISPR/CAS9 system to deselect a population of recombinant virus. Traditionally this system has been used to create a high rate of recombination, which the authors did not see. However, they were able to use this system to efficiently cleave the parental virus, and increase the selection of there recombinant. Selecting for a pure recombinant virus is the most time consuming step in this process. The authors also show the utility to make marker free virus.

The study was well throughout and well implemented. All the controls were used properly.

The discussion was encompassing of the field. I have no changes to this manuscript.

We thank this reviewer for their positive remarks.

Reviewer #2 (Remarks to the Author):

The manuscript entitled " Rapid virus engineering using CRISPR/Cas9 as a selection tool" reported repurposing Cas9 targeting as a selection tool for enriching recombinant vaccinia virus. I have the following concerns :

Major:

1. Is the selection effect of Cas9 site-specific? According to Figure 3, Cas9 reduces VACV titer regardless of the cleavage site. However, in Figure 4c and d, the copy number of mCherry decreases in response to mCh-gRNA but not A23R-gRNA, or vice versa. I believe the site effect of Cas9 targeting should be one key factor in this selection pipeline and be studied further.

We have addressed this by using a gRNA that targets an irrelevant site in the genome in Fig. 5, the supplementary video and Supplementary Fig. 3c, which supports Fig. 6. The video shows very strikingly in real time the inhibition of all virus, both the parent and the desired recombinant, when a gRNA that targets the genome away from the site of engineering is used. The scoring of progeny shows the final outcome which is reduced virus production but no selection. These new data leave no doubt that selection is gRNA specific and not an artefact of inhibiting virus replication in general.

2. The authors showed low efficiency of Cas9-mediated NHEJ repair (less than 10%) in VACV. However, in the selection process, it is possible that NHEJ mutants generated during the first Cas9 targeting could be enriched in the following steps, thus impairing the efficiency of "selection". I believe this could be a challenge especially when the initial recombination rate is low.

Our original data in Fig. 6 and Fig 7. show that this is not the case, but we did not make this point clearly in the results section of the manuscript and have made the following changes:

- In the results section describing Fig. 6, we have altered the text: "One concern with this method is the possibility of editing by NHEJ and outgrowth of these unwanted viruses during the selection steps... ..in our main experiment where were selecting

Venus⁺ virus from a background of mCh⁺ parent, there was a very low frequency of editing by NHEJ as reflected by plaques with neither fluorescent marker (double-negative) after this first selection step (Fig. 6b). The second selection step then almost completely removed the remaining parental virus, and total virus output was now similar to a non-targeted culture (Fig. 6b, Supplementary Fig. 4b). Importantly, there was no significant increase in plaques with no fluorescence, again pointing to very low levels of editing by NHEJ such that these viruses are unable to compete with the desired recombinant during selection and are not a problem.”

- In the results section describing Fig.7, we have added: “Of the 20 plaques from the two experiments, all had either parent or desired recombinant sequence, as expected from the initial PCR screen and confirmed by restriction digests. Notably none of these sequences showed evidence of editing by NHEJ, again suggesting that this process is not a significant confounder in our selection method.”

Minor:

1. If you want to claim high gene editing efficiency, one site is enough. But if you show the low efficiency, multiple sites should be tested considering that high level of gene editing has been reported earlier.

We have added panels to Figs. 1 and 2 to include a second gRNA target (eGFP) and three additional sites across the VACV genome. We have also included a new supplementary figure (S1) that shows all the sites and genes we examine in the paper and note in the discussion “the experiments shown here include gRNAs targeting five different genes spread across the VACV genome in six different locations (Supplementary Fig. 1), which strongly suggests that our data are truly representative.”

2. It is better to indicate the name of virus since only VACV was studied.

We have changed the title to “Rapid poxvirus engineering...” All poxviruses have a similar replication strategy that is exclusively in the cytoplasm so we believe that this is a reasonable generalisation.

Reviewer #3 (Remarks to the Author):

Gowripalan A. et al reported development of a method to enrich conventional poxvirus recombinants for making recombinant viruses without the aid of marker genes in less than a week. They utilized the inhibition of viral replication by Cas9 nuclease to eliminate non-recombinant population. The principle of the method is simple but I think there is no improvement or findings in this report. In addition, the use of the method is very limited because the state-of-art of selection is totally dependent on replication inhibition by Cas9, which could be seen only the vaccinia virus. The recombination rate in the infected cell or viral factory is not improved at all and considered to be remained very low.

There are some issues I concern about the experiments in the text.

1. In the video file, not all cells show fluorescence of mCherry. How this could be explained? Did authors determined infection efficiency of VACV? Seeing the video, there looks about 30% of uninfected or without mCherry fluorescent cells exist.

In the original video the amount of virus introduced at the start was low and did not spread to all the cells in the 72 hour time-frame. Our new video shows more complete infection by the end of the movie, however this does not have any bearing on our interpretation or conclusions.

2. I did not find any improvement about recombination efficiency of infected cells. It is true that the authors used “elimination strategy”, which is different from the previous reports. But I think there is no novelty in the method. Did authors find any difference or improvement in recombination efficiency in their method?

We agree that there is no improvement in recombination frequency when using CRISPR/Cas9. This is indeed the whole point of the paper and the source of our selection strategy (or “elimination strategy”), which the reviewer points out differs from previous reports. If our report differs, the finding is novel, by definition.

3. As they record time-lapse change of fluorescence from mCherry to Venus, they are able to observe and count cell numbers those have recombination events. As far as I see, the recombination events are very low and only replicated cell population is growing.

Again, this is the entire point of the manuscript – engineering poxviruses by homologous recombination is a low frequency process (with or without CRISPR/Cas). However, we can use appropriately targeted Cas9/gRNA complexes to reduce growth of parent (non-engineered) virus allowing the out-growth of the desired recombinant.

4. In experiments for Figures 6 and 7, they should use the same protocol to select recombinant-positive VACV. The efficiency of the selection is different; they get almost 100% Venus positive in Fig 6, although they only get 50-70% positive in Fig 7. How this can be explained?

Yes, we used the same protocol, however the gRNA was different because we were targeting a different place in the genome. In conventional CRISPR/Cas9 engineering, it is well known that some gRNAs work better than others and we expect that this effect is what causes the difference in efficiency seen between Fig 6 and 7.

We have added text to the discussion: “we note that the speed of selection differs across our experiments (e.g. compare Fig. 6 and Fig. 7), which may be linked to the efficiencies of the different gRNAs in directing Cas9.”

REVIEWERS' COMMENTS:

Reviewer #2 (Remarks to the Author):

The authors addressed most of my concerns and now I only have one comment: Earlier studies (PMID: 25741005 ; PMID: 26417609) that report high gene editing efficiency in vaccinia virus uses plasmid-based delivery method, however, this paper uses Cas9 ribonucleoprotein (RNP). Although Cas9 RNP has been reported to be efficient, it is relatively vulnerable to activity disruptions and has a limited working time. It is possible that different delivery methods of CRISPR/Cas9 components lead to distinct gene editing rate observed between this paper and previous ones?

Reviewer #3 (Remarks to the Author):

I think authors have properly addressed all points I have raised in my review. The revised manuscript is now suitable for publication in Communications Biology.

Response to Reviewers Comments

Comments on Revision:

Reviewer #2 (Remarks to the Author)

The authors addressed most of my concerns and now I only have one comment:

Earlier studies (PMID: 25741005 ; PMID: 26417609) that report high gene editing efficiency in vaccinia virus uses plasmid-based delivery method, however, this paper uses Cas9 ribonucleoprotein (RNP). Although Cas9 RNP has been reported to be efficient, it is relatively vulnerable to activity disruptions and has a limited working time. It is possible that different delivery methods of CRISPR/Cas9 components lead to distinct gene editing rate observed between this paper and previous ones?

All our data point to Cas9/gRNA delivered as an RNP very efficiently cutting poxvirus genomes over several days (e.g. in our supplemental movie). Further, we know of no evidence that the integrity/longevity of Cas9 impacts the activity of the complexes responsible for NHEJ or HDR – the part of the conventional CRISPR editing process that we show is inefficient in the case of poxviruses. Therefore the reviewer’s suggestion that our results are due to Cas9/gRNA delivered as an RNP having a limited working time, seems very unlikely. Having noted that, we have used plasmids as well, which was implied in the discussion where we stated that we have repeated published protocols (which includes the two references noted above). In any case, to make this even clearer, we now explicitly note our use of plasmids as well in the manuscript (lines 88,89 and 321).

Reviewer #3 (Remarks to the Author):

I think authors have properly addressed all points I have raised in my review. The revised manuscript is now suitable for publication in Communications Biology.

Thank you

Comments on Original Submission

Reviewer #1 (Remarks to the Author):

This manuscript deals with using CRISPR/CAS9 system to deselect a population of recombinant virus. Traditionally this system has been used to create a high rate of recombination, which the authors did not see. However, they were able to use this system to efficiently cleave the parental virus, and increase the selection of there recombinant. Selecting for a pure recombinant virus is the most time consuming step in this process. The authors also show the utility to make marker free virus.

The study was well throughout and well implemented. All the controls were used properly.

The discussion was encompassing of the field. I have no changes to this manuscript.

We thank this reviewer for their positive remarks.

Reviewer #2 (Remarks to the Author):

The manuscript entitled " Rapid virus engineering using CRISPR/Cas9 as a selection tool" reported repurposing Cas9 targeting as a selection tool for enriching recombinant vaccinia virus. I have the following concerns:

Major:

1. Is the selection effect of Cas9 site-specific? According to Figure 3, Cas9 reduces VACV titer regardless of the cleavage site. However, in Figure 4c and d, the copy number of mCherry decreases in response to mCh-gRNA but not A23R-gRNA, or vice versa. I believe the site effect of Cas9 targeting should be one key factor in this selection pipeline and be studied further.

We have addressed this by using a gRNA that targets an irrelevant site in the genome in Fig. 5, the supplementary video and Supplementary Fig. 3c, which supports Fig. 6. The video shows very strikingly in real time the inhibition of all virus, both the parent and the desired recombinant, when a gRNA that targets the genome away from the site of engineering is used. The scoring of progeny shows the final outcome which is reduced virus production but no selection. These new data leave no doubt that selection is gRNA specific and not an artefact of inhibiting virus replication in general.

2. The authors showed low efficiency of Cas9-mediated NHEJ repair (less than 10%) in VACV. However, in the selection process, it is possible that NHEJ mutants generated during the first Cas9 targeting could be enriched in the following steps, thus impairing the efficiency of "selection". I believe this could be a challenge especially when the initial recombination rate is low.

Our original data in Fig. 6 and Fig 7. show that this is not the case, but we did not make this point clearly in the results section of the manuscript and have made the following changes:

- In the results section describing Fig. 6, we have altered the text: "One concern with this method is the possibility of editing by NHEJ and outgrowth of these unwanted viruses during the selection steps... ..in our main experiment where were selecting

Venus⁺ virus from a background of mCh⁺ parent, there was a very low frequency of editing by NHEJ as reflected by plaques with neither fluorescent marker (double-negative) after this first selection step (Fig. 6b). The second selection step then almost completely removed the remaining parental virus, and total virus output was now similar to a non-targeted culture (Fig. 6b, Supplementary Fig. 4b). Importantly, there was no significant increase in plaques with no fluorescence, again pointing to very low levels of editing by NHEJ such that these viruses are unable to compete with the desired recombinant during selection and are not a problem.”

- In the results section describing Fig.7, we have added: “Of the 20 plaques from the two experiments, all had either parent or desired recombinant sequence, as expected from the initial PCR screen and confirmed by restriction digests. Notably none of these sequences showed evidence of editing by NHEJ, again suggesting that this process is not a significant confounder in our selection method.”

Minor:

1. If you want to claim high gene editing efficiency, one site is enough. But if you show the low efficiency, multiple sites should be tested considering that high level of gene editing has been reported earlier.

We have added panels to Figs. 1 and 2 to include a second gRNA target (eGFP) and three additional sites across the VACV genome. We have also included a new supplementary figure (S1) that shows all the sites and genes we examine in the paper and note in the discussion “the experiments shown here include gRNAs targeting five different genes spread across the VACV genome in six different locations (Supplementary Fig. 1), which strongly suggests that our data are truly representative.”

2. It is better to indicate the name of virus since only VACV was studied.

We have changed the title to “Rapid poxvirus engineering...” All poxviruses have a similar replication strategy that is exclusively in the cytoplasm so we believe that this is a reasonable generalisation.

Reviewer #3 (Remarks to the Author):

Gowripalan A. et al reported development of a method to enrich conventional poxvirus recombinants for making recombinant viruses without the aid of marker genes in less than a week. They utilized the inhibition of viral replication by Cas9 nuclease to eliminate non-recombinant population. The principle of the method is simple but I think there is no improvement or findings in this report. In addition, the use of the method is very limited because the state-of-art of selection is totally dependent on replication inhibition by Cas9, which could be seen only the vaccinia virus. The recombination rate in the infected cell or viral factory is not improved at all and considered to be remained very low.

There are some issues I concern about the experiments in the text.

1. In the video file, not all cells show fluorescence of mCherry. How this could be explained? Did authors determined infection efficiency of VACV? Seeing the video, there looks about 30% of uninfected or without mCherry fluorescent cells exist.

In the original video the amount of virus introduced at the start was low and did not spread to all the cells in the 72 hour time-frame. Our new video shows more complete infection by the end of the movie, however this does not have any bearing on our interpretation or conclusions.

2. I did not find any improvement about recombination efficiency of infected cells. It is true that the authors used “elimination strategy”, which is different from the previous reports. But I think there is no novelty in the method. Did authors find any difference or improvement in recombination efficiency in their method?

We agree that there is no improvement in recombination frequency when using CRISPR/Cas9. This is indeed the whole point of the paper and the source of our selection strategy (or “elimination strategy”), which the reviewer points out differs from previous reports. If our report differs, the finding is novel, by definition.

3. As they record time-lapse change of fluorescence from mCherry to Venus, they are able to observe and count cell numbers those have recombination events. As far as I see, the recombination events are very low and only replicated cell population is growing.

Again, this is the entire point of the manuscript – engineering poxviruses by homologous recombination is a low frequency process (with or without CRISPR/Cas). However, we can use appropriately targeted Cas9/gRNA complexes to reduce growth of parent (non-engineered) virus allowing the out-growth of the desired recombinant.

4. In experiments for Figures 6 and 7, they should use the same protocol to select recombinant-positive VACV. The efficiency of the selection is different; they get almost 100% Venus positive in Fig 6, although they only get 50-70% positive in Fig 7. How this can be explained?

Yes, we used the same protocol, however the gRNA was different because we were targeting a different place in the genome. In conventional CRISPR/Cas9 engineering, it is well known that some gRNAs work better than others and we expect that this effect is what causes the difference in efficiency seen between Fig 6 and 7.

We have added text to the discussion: “we note that the speed of selection differs across our experiments (e.g. compare Fig. 6 and Fig. 7), which may be linked to the efficiencies of the different gRNAs in directing Cas9.”